# Ixazomib, Lenalidomide and Dexamethasone in Relapsed and Refractory Multiple Myeloma in Routine Clinical Practice: Extended Follow-Up Analysis and the Results of Subsequent Therapy

**DOI:** 10.3390/cancers14205165

**Published:** 2022-10-21

**Authors:** Jiri Minarik, Jakub Radocha, Alexandra Jungova, Jan Straub, Tomas Jelinek, Tomas Pika, Ludek Pour, Petr Pavlicek, Lubica Harvanova, Lenka Pospisilova, Petra Krhovska, Denisa Novakova, Pavel Jindra, Ivan Spicka, Hana Plonkova, Martin Stork, Jaroslav Bacovsky, Vladimir Maisnar, Roman Hajek

**Affiliations:** 1Department of Hemato-Oncology, Faculty of Medicine and Dentistry, Palacky University Olomouc and University Hospital Olomouc, 779 00 Olomouc, Czech Republic; 24th Department of Internal Medicine—Hematology, Faculty Hospital, Charles University in Hradec Kralove, 500 03 Hradec Kralove, Czech Republic; 3Hematology and Oncology Department, Charles University Hospital Pilsen, 323 00 Pilsen, Czech Republic; 41st Medical Department—Clinical Department of Haematology, First Faculty of Medicine and General Teaching Hospital Charles University, 110 00 Prague, Czech Republic; 5Department of Hematooncology, University Hospital Ostrava, Faculty of Medicine University of Ostrava, 708 00 Ostrava, Czech Republic; 6Department of Internal Medicine, Hematology and Oncology, University Hospital Brno, Faculty of Medicine Masaryk University, 625 00 Brno, Czech Republic; 7Department of Internal Medicine and Hematology, 3rd Faculty of Medicine, Charles University and Faculty Hospital Kralovske Vinohrady, 100 34 Prague, Czech Republic; 8Department of Hematology and Transfusiology, University Hospital, Faculty of Medicine, Slovak Medical University and Comenius University, 831 01 Bratislava, Slovakia; 9Institute of Biostatistics and Analyses, Ltd., 602 00 Brno, Czech Republic

**Keywords:** multiple myeloma, relapsed and refractory, real-world analysis, immunomodulatory drugs, proteasome inhibitors

## Abstract

**Simple Summary:**

We report the final outomes of the addition of ixazomib to the combination of lenalidomide and dexamethasone in patients with relapsed and refractory multiple myeloma in the routine clinical practice. With prolonged follow-up, the overall response rate was similar in both cohorts, but the addition of ixazomib induced more deeper responses. Median progression free survival was significantly better in patients receiving ixazomib and translated into better overal survival. Inferior results were seen in patients who were pretreated with lenalidomide in previous regimens. We conclude that the treatment using IRD regimen in routine practice is easy, well tolerated, and with very good therapeutic outcomes, comparable to the outcomes of the clinical trial.

**Abstract:**

Background: We confirmed the benefit of addition of ixazomib to lenalidomide and dexamethasone in patients with relapsed and refractory multiple myeloma (RRMM) in unselected real-world population. We report the final analysis for overall survival (OS), second progression free survival (PFS-2), and the subanalysis of the outcomes in lenalidomide (LEN) pretreated and LEN refractory patients. Methods: We assessed 344 patients with RRMM, treated with IRD (N  =  127) or RD (N  = 217). The data were acquired from the Czech Registry of Monoclonal Gammopathies (RMG). With prolonged follow-up (median 28.5 months), we determined the new primary endpoints OS, PFS and PFS-2. Secondary endpoints included the next therapeutic approach and the survival measures in LEN pretreated and LEN refractory patients. Results: The final overall response rate (ORR) was 73.0% in the IRD cohort and 66.8% in the RD cohort. The difference in patients reaching ≥VGPR remained significant (38.1% vs. 26.3%, *p* = 0.028). Median PFS maintained significant improvement in the IRD cohort (17.5 vs. 12.5 months, *p* = 0.013) with better outcomes in patients with 1–3 prior relapses (22.3 vs. 12.7 months *p* = 0.003). In the whole cohort, median OS was for IRD vs. RD patients 40.9 vs. 27.1 months (*p* = 0.001), with further improvement within relapse 1-3 (51.7 vs. 27.8 months, *p* ˂ 0.001). The median PFS of LEN pretreated (N = 22) vs. LEN naive (N = 105) patients treated by IRD was 8.7 vs. 23.1 months (*p* = 0.001), and median OS was 13.2 vs. 51.7 months (*p* = 0.030). Most patients in both arms progressed and received further myeloma-specific therapy (63.0% in the IRD group and 53.9% in the RD group). Majority of patients received pomalidomide-based therapy or bortezomib based therapy. Significantly more patients with previous IRD vs. RD received subsequent monoclonal antibodies (daratumumab—16.3% vs. 4.3%, *p* = 0.0054; isatuximab 5.0% vs. 0.0%, *p* = 0.026) and carfilzomib (12.5 vs. 1.7%, *p* = 0.004). The median PFS-2 (progression free survival from the start of IRD/RD therapy until the second disease progression or death) was significantly longer in the IRD cohort (29.8 vs. 21.6 months, *p* = 0.016). There were no additional safety concerns in the extended follow-up. Conclusions: The IRD regimen is well tolerated, easy to administer, and with very good therapeutic outcomes. The survival measures in unsorted real-world population are comparable to the outcomes of the clinical trial. As expected, patients with LEN reatment have poorer outcomes than those who are LEN-naive. The PFS benefit of IRD vs. RD translated into significantly better PFS-2 and OS, but the outcomes must be accounted for imbalances in pretreatment group characteristics (especially younger age and stem cell transplant pretreatment), and in subsequent therapies.

## 1. Introduction

Multiple myeloma (MM) is a malignant B-cell neoplasm with heterogeneous behavior. The hallmarks of the disease include bone marrow involvement by clonal plasma cells together with the presence of monoclonal immunoglobulin (M-protein, MIG) in the serum and/or urine. The disease is associated with end-organ involvement which is mostly represented by the acronyme CRAB (C = calcium elevated, R = renal impairment, A = anemia, B = bone lesions). With the introduction of novel drugs with biological mechanism of action, the prognosis of MM has changed substantially over the last 20 years. However, most patients still relapse and require further treatment.

For patients with relapsed and refractory multiple myeloma (RRMM) we can use several therapeutic strategies, most of them combining novel agents in order to overcome drug resistance and to provide durable responses. Previously, the use of the dublet lenalidomide and dexamethasone (RD) provided a fair standard for the relapsed setting. Recently, however, several new drugs combined with RD (such as proteasome inhibitors or monoclonal antibodies) and confirmed significantly improved outcomes.

The TOURMALINE-MM1 clinical trial compared a fully-oral combination of a proteasome inhibitor (ixazomib), immunomodulatory drug (lenalidomide) and a steroid (dexamethasone) to lenalidomide and dexamethasone alone, and showed significantly better outcomes in the ixazomib arm [1]. We performed a real-world analysis that demonstrated that the IRD regimen is feasible and provides significant therapeutic outcomes including improved response rates and survival measures over lenalidomide and dexamethasone doublet (RD) even outside clinical trial setting [2].

The aim of this prospectivaly defined analysis was to assess the durability of response including the overall survival (OS), second progression free survival (PFS-2) defined as time from IRD or RD initiation until the progression on next line of therapy, or death from any cause, and the subanalysis of the outcomes in lenalidomide (LEN) pretreated and LEN refractory patients. This is the second and final part summorizing the data of the large “real-world” study.

## 2. Patients and Methods

### 2.1. Study Design

The study design of our analysis was previously described [2]. We analyzed prospectively two cohorts of RRMM patients treated solely with RD or with the addition of ixazomib within a Named Patient Program (NPP). The selection of the patients into the study was based solely on the clinicians’ preference and the availability of the NPP. The median number of previous lines was 1, still with substantial number of patients being treated after ≥4 prior lines (10.2% vs. 7.8%). We used an extended follow-up in order to define the study endpoints. For this particular analysis, we ammended the existed protocol for new primary endpoints—OS, PFS and PFS-2 to find out long term outcomes of the treatment. Secondary endpoints included the next therapeutic approach and the survival measures in LEN pretreated and LEN refractory patients. The cutoff date for the analysis was 11 March 2021.

### 2.2. Patient Population

A cohort of 344 patients with RRMM was treated with either IRD (N = 127) or RD regimen (N = 217) outside the clinical trials setting. Ixazomib was accessed via the Named Patient Program (NPP). All patients provided informed consent for participation in the study. The study was approved by institutional ethics committees, and in accordance with the Helsinki Declaration of 1975, as revised in 2008.

#### 2.2.1. Assessments

All the data were blinded and recorded in the Registry of monolonal gammopathies (RMG). The endpoints were assessed on the basis of the International Myeloma Working Group (IMWG) response criteria with the addition of minimal response criterion based upon the European Group for Blood and Marrow Transplant (EBMT) recommendations [3,4,5].

#### 2.2.2. Statistical Analysis

Depending on the nature of the data, suitable methods for description and statistical testing were selected. The variables were described with the use of absolute and relative frequencies and continuous variables using median together with min-max scale. In accordance with data continuity (categorical × continuous), Fisher’s exact test or Mann-Whitney U test was used to examine the association between selected variables and treatment regimen. Event-free survival (OS and PFS) was assessed using the Kaplan–Meier methodology. For the statistical significance of differences in survival between individual subgroups we used the log-rank test.

The statistical analysis was performed at a significance level of α = 0.05 (tests two-sided). For the anylysis we used the SPSS software (IBM Corp. Released 2016. IBM SPSS Statistics for Windows, Version 24.0.0.1 Armonk, NY, USA: IBM Corp.) and with the help of software R version 3.4.2. [6].

## 3. Results

### 3.1. Patients and Baseline Characteristics

The baseline demographics and characteristics were described previously [2]. Apart from minor statistically significant differences, the cohorts were well balanced. The IRD cohort had slightly lower median age than the RD cohort (66.0 vs. 68.0 years, *p* = 0.002), a higher proportion of patients underwent autologous stem cell transplant (ASCT) (62.0% vs. 43.3%, *p* ˂ 0.001), and slightly higher percentage were pretreated by proteasome inhibitor (96.9% vs. 91.2%, *p* = 0.047). Also, the IRD cohort had higher proportion of extramedullary myeloma (EM) than the RD cohort (14.2% vs. 6.7%, *p* = 0.034). Finally, the starting dose of lenalidomide (LEN) was slightly but significantly higher in the IRD cohort: LEN 25 mg (72.8% vs. 63.3%), LEN 15 mg (20.8% vs. 17.2%), LEN 10 mg (6.4% vs. 16.7%), and LEN 5 mg (0% vs. 2.2%). The univariable and multivariable analyses showed no impact of uneven baseline variables on response rates or survival measures. All other variables were without any significant differences between the cohorts.

The median follow-up was 28.5 months in all patients, 31.7 months in the IRD cohort and 26.0 months in the RD cohort.

### 3.2. Response Rates

The response rates remained similar to those reported previously with slight improvement over time. The final overall response rate (ORR) was 73.0% in patients treated with IRD vs. 66.8% in patients treated with RD regimen. The maximum treatment response rates are summarized in Table 1: complete response (CR) in 11.9% vs. 7.8%, very good partial response (VGPR) in 23.8% vs. 18.4%, partial response (PR) in 34.9% vs. 40.6%, and minimal response (MR) in 9.5 vs. 15.2%. The difference in patients reaching ≥ VGPR remained significant (38.1% vs. 26.3%, *p* = 0.028).

### 3.3. Progression Free Survival

Median PFS was significantly improved in the IRD cohort (17.5 vs. 12.5 months, *p* = 0.013). Patients with 1–3 prior relapses treated with IRD had median PFS improved, too (22.3 vs. 12.7 months *p* = 0.003), Figure 1. The results were similar to our previously reported analysis [2].

The best results of IRD regimen were seen in patients in the first relapse with decreasing median PFS with the second and third relapse of the disease (30.1 vs. 20.0 vs. 9.7 months). In the RD cohorts, the median PFS was decreasing with later relapses, too (15.2 vs. 9.3 vs. 9.2 months), Figure 2.

### 3.4. Overall Survival

Median OS was significantly better in patients receiving IRD. In the whole cohort, median OS was 40.9 vs. 27.1 months (*p* = 0.001), with further improvement within relapse 1–3 (51.7 vs. 27.8 months, *p* ˂ 0.001), Figure 3.

Median OS in the IRD cohort also confirmed better outcomes in the first and second relapse (51.7 months and not reached) versus the third relapse (13.2 months). The outcomes in the RD cohort were with decreasing survival with later relapses, too (35.9 vs. 24.4 vs. 15.6 months, relapse 1–3, respectively), Figure 4.

### 3.5. Lenalidomide Pretreatment and Refracterity

Patients pretreated with lenalidomide in any previous line of therapy had inferior outcomes. In the IRD cohort, the median PFS of LEN pretreated (N = 22) vs. LEN naive (N = 105) patients was 8.7 vs. 23.1 months (*p* = 0.001), and median OS was 13.2 vs. 51.7 months (*p* = 0.030), Figure 5. In the RD cohort, there was a trend towards worse outcomes in LEN pretreated (N = 33) vs. LEN naive (N = 184) patients, but the results were not significant (mPFS 9.2 vs. 13.2 months, mOS = 23.0 vs. 28.5 months, *p* = NS). There were 7 LEN refractory patients in the IRD cohort and 5 LEN refractory patients in the RD cohort—their outcomes remained poor regarding both median PFS (8.7 and 3.8 months) and median OS (8.7 and 5.3 months, respectively).

### 3.6. Next Treatment

With extended follow-up, most patients in both cohorts progressed or died, and most of them received further myeloma-specific therapy (63.0% in the IRD group and 53.9% in the RD group), Figure 6. Majority of patients received pomalidomide-based therapy or bortezomib based therapy. The representation was similar in both cohorts. Other therapeutic modalities were used less frequently and with differences among the cohorts. Interestingly, significant number (up to 18%) of patients received LEN based retreatment (in most cases LEN-based triplets such as carfilzomib-lenalidomid-dexamethasone, daratumumab-lenalidomid-dexamethasone, elotuzumab-lenalidomid-dexamethasone or ixazomib-lenalidomide-dexamethasone). 

Monoclonal antibodies were used more frequently in patients with previous IRD vs. RD treatment (daratumumab—16.3% vs. 4.3%, *p* = 0.0054; isatuximab 5.0% vs. 0.0%, *p* = 0.026). Similarly, significantly more patients with previous IRD vs. RD received subsequent carfilzomib (12.5 vs. 1.7%, *p* = 0.004). Patients relapsing from RD were more frequently treated with ixazomib (6.8%), nivolumab (2.6%) or ibrutinib (1.7%) but the differences were not significant.

Median PFS in the next treatment line in all relapsing patients was 6.9 months and median OS was 16.0 months (Figure 6). There were no significant differences in either group regarding median PFS (IRD vs. RD 6.2 vs. 7.8 months, *p* = 0.563) or median OS (IRD vs. RD 17.2 vs. 15.1 months, *p* = 0.345). The median PFS-2 (progression free survival from the start of IRD/RD therapy until the second disease progression or death) was significantly longer in the IRD cohort (29.8 vs. 21.6 months, *p* = 0.016), Figure 7.

### 3.7. PFS 2-Progression-Free Survival from Date of Initiation Line of Therapy with IRD/RD to Disease Progression on Next Line of Therapy, or Death from Any Cause, Whichever Occurs First

As there were many different combination regimens with small number of patients, we did not assess for outcomes in individual combinations in the next treatment line.

### 3.8. Safety

There were no additional safety concerns in the extended follow-up. Majority of grade ≥3 adverse events (AEs) included hematological toxicity (anemia, neutropenia, thrombocytopenia), and there were no significant differences among the cohorts. Patients receiving IRD had significantly more grade 1–2 infections (51.0% vs. 36.5%, *p* = 0.029)*,* the rate of grade ≥3 infection was not different (21.6% vs. 21.2%).

More cases of peripheral neuropathy (PN) were seen in the IRD cohort—grade 1 PN was present in 35.2% vs. 32.4%, grade 2 in 14.8% vs. 8.7%, grade 3 in 4.6% vs. 1.2%, (*p* = 0.044). Finally, diarrhea was more frequent in patients receiving ixazomib (all grades in 34.7% vs. 19.5%, *p* = 0.026) but only 1 patient had grade ≥3 diarrhea.

Only 7.2% patients in the IRD cohort (8/127) and 8.2% in the RD cohort (15/217) discontinued the treatment due to toxicity.

## 4. Discussion

Over the past 20 years the MM treatment landscape has greatly expanded, and the progress continues at a rapid pace. Many novel therapies with biological mechanism of action have been introduced, leading to deeper therapeutic responses and prolonged survival of MM patients [7]. Clinical trials have brought evidence on the increasing outcomes in both, newly diagnosed (NDMM) and relapsed or refractory multiple myeloma patients (RRMM). However, it has been reported that up to 40% of NDMM patients and over 70% of RRMM patients treated in routine practice would not have been eligible for participation in clinical trials based on their inclusion and exclusion criteria [8,9,10]. Clinical trials usually use a select cohort of patients who are healthier than the general population, with better performance status, with slower progression of the disease and with better estimated survival [8]. Within the clinical trials, there is an under-representation of elderly and co-morbid patients and patients from lower socio-economic background [8,11,12]. Additionally, the difference in the approaches to treatment in academic versus community centers contributes to the fact that the data from clinical trials are not always generalizable to the wider population.

Therefore, the real-world evidence (RWE) expediently supports the evidence from clinical trials. RWE relates to patient health, therapeutic effectiveness, safety and delivery of healthcare and collects data from local, national, or global clinical registries, electronic medical records, and databases of billing or insurance claims [13]. Of course, RWE works with heterogeneous datasets from different sources, with lack of control mechanisms, with no randomization or blinding and the data from RWE are thus susceptible to selection bias [14,15]. RWE is not meant to replace clinical trials but rather to provide additional insights in a broader population of patients, and in helping us understand how the improvements reported by clinical trials translate to routine clinical practice [14].

Several registries are currently aiming to prospectively collect evidence about real-world outcomes in MM to better understand how treatments may be personalized to achieve optimal outcomes for MM patients [7]. The presented data were acquired from the Czech Registry of Monoclonal Gammopathies (RMG), which is an international registry specifically aimed at clinical data on the characteristics of MM patients, covering predominantly central Europe [16].

The treatment with IRD combination demonstrated the longest median OS data to date in RRMM phase III studies treated with lenalidomide based triplets [17]. The median OS was 53.6 vs. 51.6 months in favor of the IRD regimen, still the difference was not statistically significant, possibly confounded by the imbalances in subsequent therapies. Patients with adverse prognostic factors yielded greater OS benefit. Statistically significant benefit in OS was seen in the China continuation study, which, however, had only 50% of patients receiving subsequent therapy with narrower spectrum of approved or investigational agents than in patients from North America or Europe [17,18]. The final analysis of the Tourmaline-MM1 trial therefore demonstrated two important outcomes: First, that the IRD combination may be still considered a useful treatment option for RRMM, having strong survival data, being easy to administer and with low additional toxicity versus placebo-RD. Second, that there is an increasing role of novel subsequent therapies and their sequencing for the OS in MM patients.

Several RWE worldwide supported the benefit of ixazomib-based treatment in RRMM patients regarding PFS [19,20,21,22,23,24,25,26]. None of these studies compared IRD vs. RD but the outcomes were quite similar with median PFS varying from 17–31 months. Shorter mPFS were reported in Hungarian study, possibly due to more heavily pretreated patients and LEN supply issues, similarly the Canadian MCRN analysis and Chinese real-world analysis reported shorter mPFS, possibly due to lenalidomide or bortezomib pretreatment and due to inclusion of other ixazomib based therapies [24,25,26]. Similarly, in the retrospective analysis by Takakuwa et al., the outcomes of IRD were not favorable, probably due to previous pretreatment (with median of four prior therapies) including lenalidomide and bortezomib [27]. On the other hand, the Slovak data from routine practice demonstrated very long PFS of 43 months [28]. Longer PFS of 27.6 months was demonstrated in the study of Terpos et al., too [20].

Our study can only supplement the RWE for the Tourmaline-MM1 trial. Due to the specific conditions in the Czech Republic at the time of the analysis, we were able to compare similar cohorts of patients being treated by either IRD or RD. At the time of the analysis, there was no other lenalidomide based triplet reimbursed and there was no competing clinical trial. Being a part of a Named Patient Program (NPP), ixazomib was distributed for free to the patients—which eliminated possible economical burden of the drug combination. Finally, the gradual acceptance of the NPP in eight hematological centres together with unified therapeutic approaches based on national guidelines and RMG database enabled a unique comparison of two therapeutic approaches based solely on the availability of ixazomib. Nevertheless, apart from the nature of the RWE, there were some possible sources of bias including the differences in patients´age, pretreatment by PI and ASCT, and in the dose of lenalidomide [2].

In this follow-up analysis we updated the outcomes of the two regimens, but we also aimed at specific characteristics of our cohort—especially at the presence of lenalidomide-refractory population, that would not be eligible for the clinical trial, and also at the OS and outcomes of the next treatment after IRD or RD therapy. As expected, LEN pretreated and LEN refractory patients had poorer outcomes in comparison with LEN naive patients, the median PFS did not exceed 10 months in any of the treatment arms but the patient count was too small in both arms to make any valid conclusions.

The median OS remained significantly better in the IRD vs. RD cohort (40.9 vs. 27.1 months), and the subanalysis of patients in relapse 1–3 (i.e., the population recruited by the Tourmaline-MM1 trial) demonstrated further improvement (mOS 51.7 vs. 27.8 months). The outcomes of the RWE in the IRD arm strongly correspond to the results reported by the clinical trial.

In comparison to our previous analyses, we recorded 1% less patients achieving CR in the RD arm which was due to regular monitoring of the RMG registry that revealed missing data on urine immunofixation analysis—the patients were therefore newly assessed as having VGPR. In most patients, the final response increased per time and the final outcomes are therefore better in comparison to our previously reported data [2,29,30,31].

In accord with the opinion of Richardson et al., we expect that our results for OS were largely confounded by subsequent therapies [17]. Patients relapsing from IRD were more frequently treated using monoclonal antibodies and carfilzomib. We hypothesize that the reason was due to slightly but significantly younger age of the patients in IRD vs. RD arm (median age at treatment initiation 66.0 vs. 68.0 years) which turned into possibly more fragile population after relapse. Nevertheless, the median PFS in the next treatment line was similar in both cohorts which in turn translated into maintained significant difference in PFS2, defined as progression free survival from the start of IRD/RD therapy until the second disease progression or death (29.8 vs. 21.6 months), in favor of the ixazomib-based therapy.

As more patients have finished the treatment, we were able to collect the safety data in both of the cohorts. There were no additional safety concerns, demonstrating that IRD combination is well tolerated even within an unsorted population.

## 5. Conclusions

We conclude that the IRD regimen is well tolerated, easy to administer, and with very good therapeutic outcomes. The survival measures in unsorted real-world population are comparable to the outcomes of the clinical trial. As expected, patients with LEN reatment have poorer outcomes than those who are LEN-naive. The broad use of novel agents in late phase MM leads to less beneficial outcomes than in less pretreated patients, still, it is able to further improve the survival measures of MM patients.

## Figures and Tables

**Figure 1 cancers-14-05165-f001:**
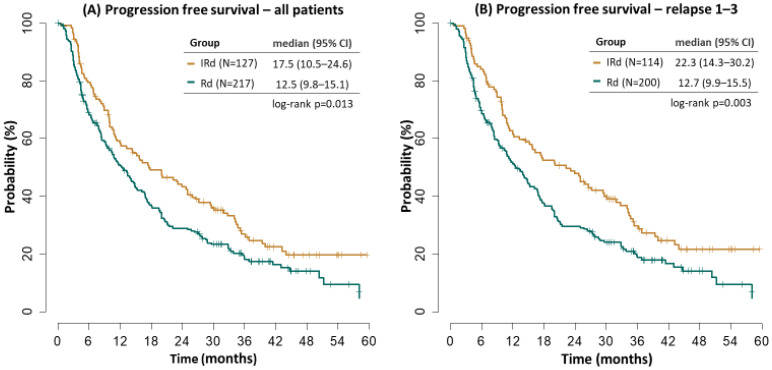
Progression free survival.

**Figure 2 cancers-14-05165-f002:**
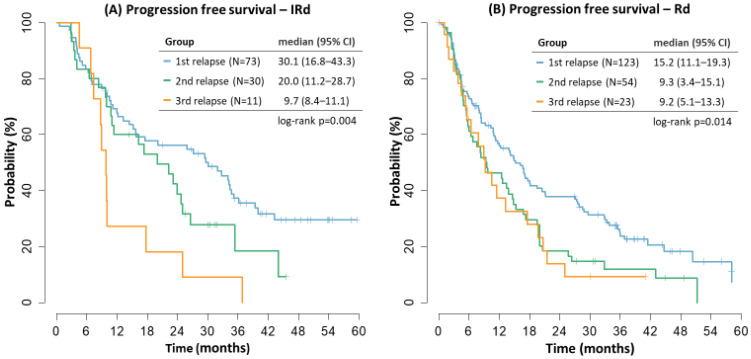
Progression free survival by treatment line.

**Figure 3 cancers-14-05165-f003:**
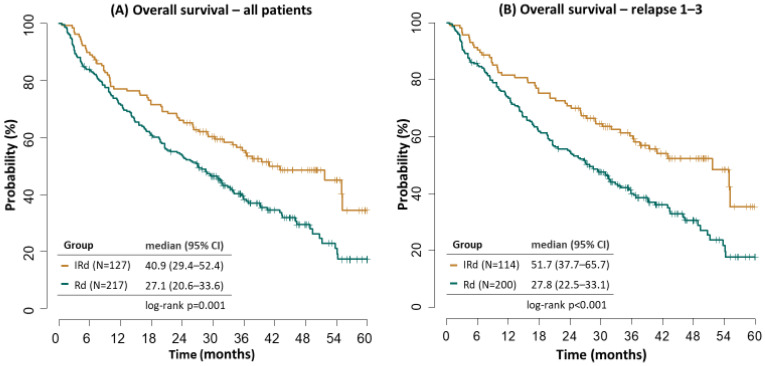
Overall survival.

**Figure 4 cancers-14-05165-f004:**
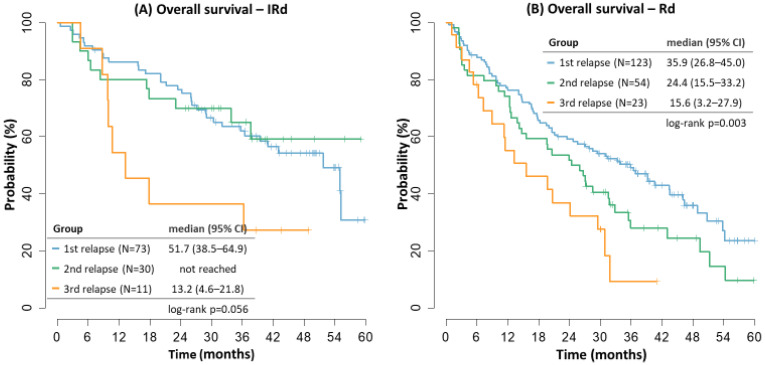
Overall survival by treatment line.

**Figure 5 cancers-14-05165-f005:**
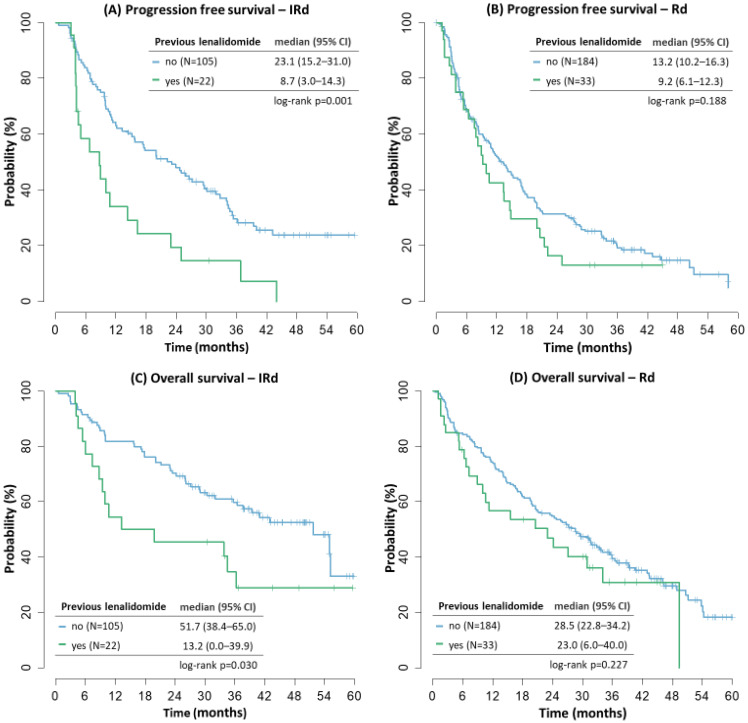
Progression free survival and overall survival by lenalidomide pretreatment.

**Figure 6 cancers-14-05165-f006:**
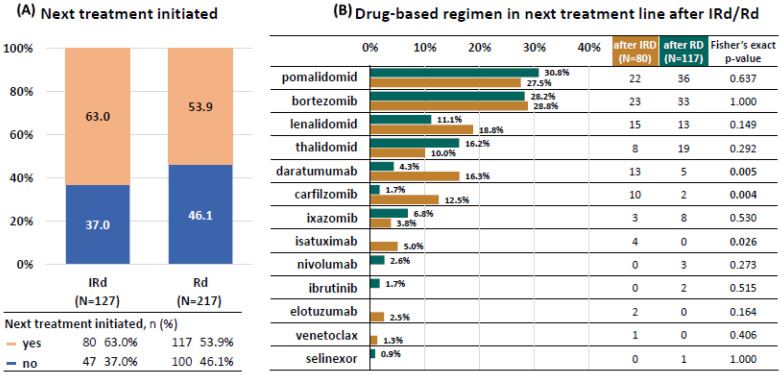
Next treatment after IRD/RD.

**Figure 7 cancers-14-05165-f007:**
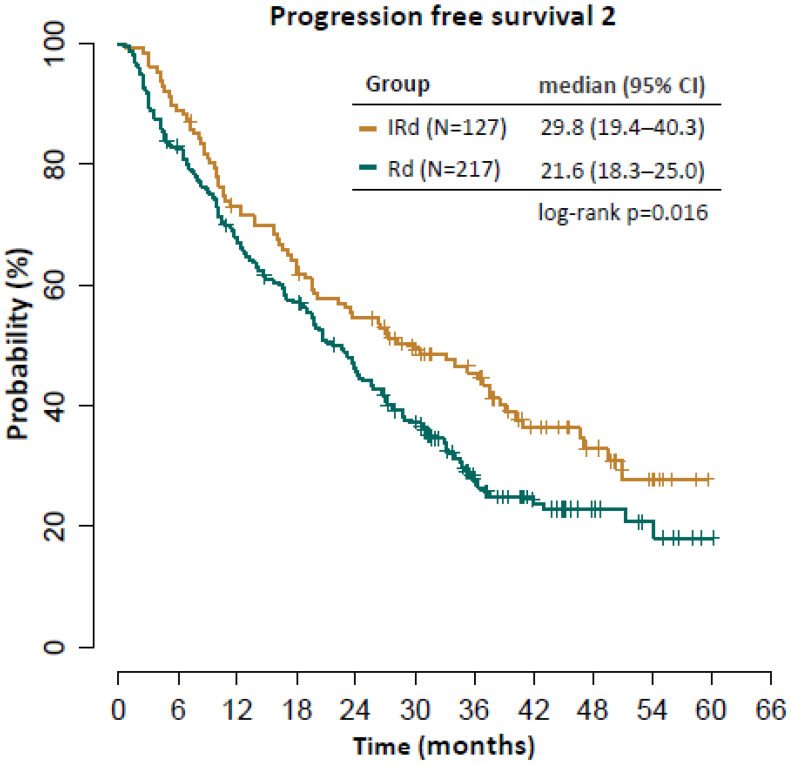
Progression free survival 2.

**Table 1 cancers-14-05165-t001:** Maximal treatment response.

	IRD (N = 127)	RD (N = 217)	*p*-Value ^1^
Maximal Treatment Response, n (%)	n = 126	n = 217	
sCR	3 (2.4%)	–	0.077
CR	15 (11.9%)	17 (7.8%)	
VGPR	30 (23.8%)	40 (18.4%)	
PR	44 (34.9%)	88 (40.6%)	
MR	12 (9.5%)	33 (15.2%)	
SD	13 (10.3%)	17 (7.8%)	
PD	9 (7.1%)	22 (10.1%)	
VGPR+ ^2^	48 (38.1%)	57 (26.3%)	**0.028**
ORR ^3^	92 (73.0%)	145 (66.8%)	0.275
CBR ^4^	104 (82.5%)	178 (82.0%)	1.000

^1^*p*-value of Fisher´s exact test. ^2^ VGPR+—patients reaching at least very good partial response. ^3^ ORR—Overall Response Rate (PR or better). ^4^ CBR—Clinical Benefit Rate (MR or better).

## Data Availability

The data of each patient were blinded and recorded under a unique code into the Registry of Monoclonal Gammopathies (RMG) of the Czech Myeloma Group (CMG). The data presented in this study are available on request from the corresponding author. The data are not publicly available due to local restrictions as well as GDPR.

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
