# Peer review of "Ixazomib, Lenalidomide and Dexamethasone in Relapsed and Refractory Multiple Myeloma in Routine Clinical Practice: Extended Follow-Up Analysis and the Results of Subsequent Therapy"

_cancers, 2022, doi:10.3390/cancers14205165_

Round 1
Reviewer 1 Report
Minarik et al. present a very well written and easy to follow report on real work data of response and survival of Ixazomib, Revlimid and Dex compared to Revlimid and Dex using real world data. The methods and analysis are very solid, data is nicely presented and the conclusions are supported by the presented results. Overall this is a very nice paper. The reviewer has only few concerns:
- Please add one or two sentences in the Study design section to explain the study design. While the authors refer to a previous study, it would be helpful to mention real world data and prospective analysis in this paragraph. Also, were patients in both groups matched or was everybody from the registry included in this study?
- what was the median lines of therapies for IRD vs RD? Please add this info to the text.
- Please elaborate why PFS2 was chosen as an endpoint . It appears that this endpoint is only a reflection from PFS of IRD vs RD treatment as the authors mention that PFS of subsequent treatment between these groups did not differ.
-Please add percentage of patients with diarrhea in IRD vs RD group into the "safety" paragraph and add the p value.
- in Table 1, please superscript numbers 2,3,4
Author Response
Thank You very much for the review and especially for all the edits. Here is the list of our actions taken:
We included two sentences that sum up the design of the study. Some information explaining the design is in the discussion section, too.
The patients were not matched – as this information is clear from the study population and is defined in the former paper, we did not include it in this paper again.
We added a sentence on the median of previous lines and also the information about patients being treated above the 4th line (i.e. outside the criteria of the clinical trial).
We added the inforation why OS and PFS2 were chosen. Yes, the next treatment did not add any substantial PFS difference which supports the idea of long term benefit of IRD regimen. However, being a „real world“ study with quite heterogeneous treatment patterns after RD/IRD, we cannot derive straightforward conclusions (especially as OS had significant difference in our study but not in the randomized trial). This is all mentioned within the discussion.
We added the percentages of diarrhea including the p value.
We corrected the superscript in Table 1.
Reviewer 2 Report
1. It would be helpful that authors specifically mention in the introduction/Abstract that this is the second and final part of the broad study.
2. It would also be helpful for the general audience if authors provide any hypothesis/thoughts about the mechanistic aspect of this improved patient outcome with inclusion of Ixazomib in the therapeutic regimen.
3. The studies cited in this research article have been systematically put forward and the outcome of all the studies have been well summarized.
4. Scientifically correct conclusions have been drawn while evaluating the statistical data.
5. The table included in the paper provides an opportunity to have a quick and concise look at the response rates vs different therapeutic modalities.
6. The research article has been neatly written with acronyms and abbreviations properly explained for greater accessibility.
7. Overall the research article is a good read which provides critical data relevant to the topic.
Author Response
Thank You very much for Your review and for very kind comments. Here are our minor changes that hopefully will support Your suggested edits:
We added the information about the second and final part of the study in the introduction section.
We added the rationale for addition of ixazomib to the RD combination in the introduction section. As we expect majority of the audience being involved in hemato-oncology (with well known mechanisms of action of individual drugs), we did not go further into the details of the synergistic action of IMiDs and proteasome inhibitors.
Reviewer 3 Report
This manuscript compares the efficacity of ixazomib, lenalidomide and dexamethasone (IRD) versus lenalidomide and dexamethasone (RD) in patients with relapsed and refractory multiple myeloma (RRMM), in terms of Overall Survival (OS).
Previous studies, including some by the same authors, have reported the benefit of ixazomib-based treatments in RRMM patients regarding PFS. In this new study OS is analyzed and the results seem to confirm this benefit.
This study could be completed by analyzing the possible correlation between cytogenetic alterations and the response to IRd or Rd treatment.
On the other hand, other studies in the real world have not shown a benefit of including ixazomib in the treatment of MM. For instance, in the study by Takakuwa et al IRd showed poor efficacy, but this work is not discussed in the manuscript. Also, studies comparing the efficacy of IRd and VRd should be discussed.
Author Response
Thank You very much for your thorough review. We completely agree with your suggestions, still due to severeal limitations were not able to realize all of the suggested improvements. Here is the list of our responses and changes made in the text:
Yes, we agree, that cytogenetics is of an importance - especially as the IRD regimen was found to improve the outcomes of patients with adverse cytogenetics in the Tourmaline-MM1 trial. However, being a real-world population, we did not have sufficient data from all the patients. Most of our patients had cytogenetics assessed – but only at the time of diagnosis and not at relapse (unlike the original trial). Moreover, there were only a few patients carrying adverse cytogenetic changes in our cohort which precluded valid statistics. These information are already mentioned in our previous paper. If possible we prefer not to repeat them in this paper as there is no new information and the outcomes are not as solid as for the primary and secondary endopints.
The study of Takakuwa et al does demonstrate the limited efficacy of IRD but not in comparison with RD. On the other hand it clearly shows that in real-world practice the spectrum of patients is different. Most of the patients were highly pretreated (with median 4 previous linies) and majority with bortezomib as well as lenalidomide pretreatment which might have affected the outcomes. But we agree that this study is relevant and we included the information as well as citation within the discussion.
We did not include the studies comparing VRD and IRD (or KRD) as they are largery retrospective and with heterogeneity in the study population. The major focus on the limitations in clinical trials (which is one of the conclusions of such trials) is discussed in the text, too.